# Interventions Based on Mind–Body Therapies for the Improvement of Attention-Deficit/Hyperactivity Disorder Symptoms in Youth: A Systematic Review

**DOI:** 10.3390/medicina55070325

**Published:** 2019-06-30

**Authors:** Yaira Barranco-Ruiz, Bingen Esturo Etxabe, Robinson Ramírez-Vélez, Emilio Villa-González

**Affiliations:** 1Department of Physical and Sports Education, PROFITH “PROmoting FITness and Health through Physical Activity” Research Group, Sport and Health University Research Institute (iMUDS), Faculty of Education and Sport Sciences, University of Granada, 52005 Melilla, Spain; 2Department of Health Sciences, Public University of Navarra, Navarrabiomed-IdiSNA, Complejo Hospitalario de Navarra (CHN), 31008 Pamplona, Spain

**Keywords:** mind–body therapies, relaxation therapy, attention-deficit/hyperactivity disorder symptom, children, adolescents

## Abstract

*Background and objectives*: Attention-deficit/hyperactivity disorder (ADHD) is one of the most common psychiatric disorders in children and adolescents. Mind–body therapies (MBTs) seem to be effective for improving health in different populations; however, whether a positive effect occurs in children and adolescents with ADHD is still controversial. The main aim of this systematic review was to analyse the interventions based on MBT aimed to improve the main ADHD symptoms in children and adolescents. *Materials and Methods*: A systematic review was conducted following the preferred reporting items for systematic reviews and meta-analyses (PRISMA) guidelines to identify MBT studies on children and adolescents (4–18 years) with a clinical diagnosis of ADHD. Study quality was evaluated by the NIH quality tool (U.S. National Institute of Health). *Results*: There were positive results in eleven out of twelve included studies regarding the effect of the MBT interventions on ADHD symptoms. With respect to ADHD symptoms, we observed differences across studies. In relation to the studies’ quality, eleven studies were rated “poor” and one was rated as “fair”. *Conclusions*: MBTs, such as yoga or mindfulness, could be positive strategies to mitigate ADHD symptoms in children and adolescents. However, further research with high-quality designs, with randomization, greater sample sizes, and more intensive supervised practice programs are needed.

## 1. Introduction

Attention-deficit/hyperactivity disorder (ADHD) is a common behavioral disorder that affects from 1% to 20% of the children worldwide [1,2], but overall, the pooled prevalence of ADHD is around 5% [3]. ADHD is characterized by pervasive and impairing symptoms of inattention, hyperactivity, and impulsivity according to the Diagnostic and Statistical Manual of Mental Disorders, fifth edition (DSM-5) [4]. The etiology of ADHD is multifaceted and multidimensional, linking genetic and environmental factors [5]. ADHD has been characterized as a major public health problem due to the broad range of negative effects in diagnosed people and the serious expenses for families and society. According to the American Psychiatric Association in the DSM-V, to be considered ADHD a child must have symptoms before the age of 12 years, for at least six months, and affecting two domains of life [6]. The prevalence of the three subtypes of ADHD are: inattentive (20–30% of diagnosed population), hyperactive-impulsive (less than 15%), and combined subtype (50–75%) [7]. Eleven percent of children ages 4 to 17 in the United States (6.4 million children) have been diagnosed with attention-deficit hyperactivity disorder, according to a survey by the Centers for Disease Control and Prevention [8]. Boys are diagnosed with ADHD three times more frequently than girls. Therefore, evidence corroborates the solid influence of genetic factors on the expression of symptoms; however, psychosocial, environmental, and cultural factors also play an important role in this disorder [9].

To date, ADHD symptoms have been treated with different therapies. A previous systematic review study suggested aerobic exercise had a moderate to large influence on core symptoms of ADHD in children and adolescents [10]. This study indicated that short-term aerobic exercise attenuates ADHD symptoms such as attention, hyperactivity, impulsivity, anxiety, executive functions, and social disorders in diagnosed children. In a study with a sample of 7000 youths with mental health concerns, such as ADHD, depression, and anxiety, complementary alternative medicine therapies had an important role, where mind–body therapies (MBTs) were the most used [11]. Accordingly, MBTs such as yoga, mindfulness, or meditation offered a wide range of positive effects on psychosocial, emotional, and neurobiological functioning in several populations [6]. These non-invasive techniques are based on positive thoughts and emotions to improve emotional and behavioral health. Such techniques consist of deep breathing, meditation, mindfulness, guided imagery, progressive relaxation, and yoga exercises [12]. It has been shown that the practice of these therapies can change brain activation patterns, contribute to improve mood and reduce anxiety, stress, and pain, in addition to developing attention control and inhibitory skills [13]. The neurophysiological mechanisms implicated respond to activation to dopaminergic and noradrenergic systems. The successful pharmacological treatments used from preschooler ages are mainly composed of psychostimulant drugs that increase dopaminergic action [14], however, the MBTs could active the liberation of this hormone without the typical adverse side effect of these chemical medications [15]. In light of these benefits, a few intervention studies have investigated the function of MBTs in controlling ADHD symptoms in young people. However, the potential benefits of MBTs on symptoms of ADHD in youths are still controversial. Thus, the main aim of this systematic review (SR) was to analyse the interventions based on MBT aimed to improve the main ADHD symptoms in children and adolescents.

## 2. Materials and Methods

We followed the recommendations of preferred reporting items for systematic reviews and meta-analyses (PRISMA) statement [16]. The PRISMA checklist of this SR can be found in Appendix A.

### 2.1. Study Eligibility Criteria

Intervention studies focused on MBTs carried out to improve the main symptoms of ADHD in children and adolescents (4–18 years old) were included in the current systematic review. First, the current study included studies enrolling children and adolescents with a clinical diagnosis of ADHD. All ADHD subtypes (i.e., combined type, predominantly inattentive, and predominantly hyperactive/impulsive) were considered as valid studies, following a previous search protocol [10]. Studies including patients with comorbid conditions (such as anxiety, depression, epilepsy, or other medical conditions) were not eligible. Second, the present review considered studies evaluating only MBT-based interventions such as, yoga, meditation, mindfulness, relaxation, or zen therapies or programs. Finally, the main outcomes of the studies included in this SR were the principal ADHD symptoms, such as inattention, hyperactivity, or impulsivity, and related skills, which are attenuated with ADHD diagnostics (e.g., accuracy rate, reaction time, etc.).

Other secondary outcomes related to ADHD, such as anxiety, shyness, social problems, perfectionism, self-reported emotion dysregulation, depressive symptoms, and parent’s emotional regulation and feelings, were included in the description of the intervention studies.

### 2.2. Search Methods for Identification of Studies

A PICOS approach was used for framing the research question and the evidence search [17]. Participants: *Child* OR adolescent* OR young* OR youth*. Interventions: *Yoga OR yogic OR meditation OR Tai Chi OR mindfulness OR mindful OR mindfulness-based OR mind-body OR relaxation OR zen*. Comparisons: Not applicable. Outcomes: Medical condition terms (*ADHD OR attention deficit OR attention-deficit OR hyperkinetic syndrome OR hyperkinetic disorder*). Study design: *intervention OR program OR therapy OR training OR school-based intervention*. Additional filters: All database, builder terms: Title/abstract for PubMed, Topic for Web of Science, and abstract for PsycINFO and EBSCOHost, Publication years from 2000 to 2018.

Thus, relevant randomized trials included in systematic reviews were reviewed. A search was performed in the following databases with no language/date/type of document restrictions: Web of Knowledge databases (Web of Science (science citation index expanded), PubMed, PsycINFO, and EBSCOHost, in order to identify additional relevant intervention studies published beyond these reviews [18]. The specific electronic search for each database and terms included are shown in Appendix A.

Additionally, references from the included studies were hand-searched to discover any potential study not detected with the electronic search.

### 2.3. Screening and Data Extraction

#### 2.3.1. Screening

The eligibility process was conducted in two separate stages:The authors (E.V.-G. and B.E.E.) independently screened titles and abstracts of all nonduplicated papers and excluded those with exclusion criteria. A definitive list was established. Discrepancies were resolved by consensus between the authors. When there was no consensus, a third, senior author (Y.B.-R.) acted as mediator. If any doubt about inclusion existed, the article proceeded to the next stage.Those articles that passed screening were downloaded (full text) and assessed for eligibility by two authors (E.V.-G. and B.E.E.), independently. Again, discrepancies were resolved by consensus between the authors, and if needed, two seniors author (Y.B.-R. and R.R.-V.) acted as arbitrators. When required, the corresponding authors of screened studies were contacted to inquire about study eligibility (n = 1).

Duplicates, nonintervention studies, non-English language, and studies without analysis of our primary outcomes or main participants were eliminated. Articles included in the review are showed in the PRISMA flow diagram (Figure 1).

#### 2.3.2. Data Extraction

Data extraction was independently performed by two researchers (E.V.-G. and Y.B.-R.); discrepancies were resolved by agreement between the two authors. The following data were extracted:Publication details: year of publication and country where the study was conducted.Study participants’ details: sample and age range.Design of the study.Intervention characteristic: duration and frequency of the intervention period.Outcome: main outcome, secondary outcomes, and assessment test.Main study results.

### 2.4. Study Quality Assessment

Study quality was evaluated using the quality assessment tools from the Risk Assessment Workgroup of the Department of Health and Human Services from the U.S. National Institute of Health [19]. We assessed quality by two different tools: (1) quality assessment of controlled intervention studies (i.e., randomized controlled trials (RCTs) and clinical trials studies), and (2) quality assessment tool for before–after (pre–post) studies with no control group. These instruments were created to evaluate the internal validity of a trial, the extent to which the reported effects can strictly be attributed to the intervention applied, and the potential flaws in methodology or implementation. Each tool contains specific questions to assess bias, confounders, power, and strength of association between intervention and outcomes. The answer to each question could be “yes,” “no,” “cannot determine,” “not reported,” or “not applicable”. A numeric scoring system was not used. The evaluator had to consider the potential risk for bias in the study design whenever a “no” was selected. Overall quality ratings were scored as “good” (low risk of bias, valid results), “fair” (some risk of bias, does not invalidate results), or “poor” (significant risk for bias, may invalidate results). If a study had a “fatal flaw” then risk of bias was significant and the study was of poor quality. Examples of fatal flaws in RCTs include high dropout rates, high differential dropout rates, intention to treat analysis, or other inappropriate statistical analysis. All studies were independently screened by one author (Y.B.-R.), and three additional reviewers (E.V.-G., B.E.E., R.R.-V.) tested a 50% sample (n = 6) to double check for accuracy.

## 3. Results

### 3.1. Study Selection

In the first stage of the search strategy, a total of (n = 506) articles were identified. In the second stage, following the removal of duplicates (n = 157), a total of (n = 349) articles were screened by title/abstract. Then, 316 potentially relevant articles were excluded with reasons (details summarized in Figure 1). In the third stage, full-text articles were reviewed in depth and (n = 21) studies were excluded with reasons (details summarized in Figure 1). Finally, only (n = 12) studies based on MBT interventions met the inclusion and exclusion criteria and were included in the final analysis.

### 3.2. Characteristics of the Included Studies

The characteristics of the included studies are presented in Table 1. The included studies were conducted in different continents, such as “the Americas” [20,21], Europe [22,23], Asia [24,25,26,27,28,29,30], and Australia [31]. All analyzed studies targeted children and adolescents (ranged from 5 to 18 years old). Only one study [20] also included adults, but results from adolescents and adults were displayed separately. Moreover, most of the studies (11/12) [20,21,22,23,24,25,27,28,29,30,31] included parental participation, supporting children to perform the program at home or being involved in the intervention. The number of participants varied across the intervention studies, ranging from eight participants [20] to 100 [28]. Furthermore, the range of parent participation also varied from 11 [29] to 79 [25] families. Regarding the study design, there were five RCT studies [25,27,28,30,31] and a clinical trial [24].

### 3.3. Analysis of the MBT Intervention and its Effects on ADHD Outcomes

#### 3.3.1. MBT Interventions

All interventions consisted of MBTs, including nine studies developing mindfulness treatment [20,21,22,23,25,27,28,29,30] and three studies that conducted yoga [24,26,31]. Five studies [20,24,26,29,31] declared that interventions were guided by a certified instructor in mindfulness or yoga. For instance, in the study of [20] instruction was carried out by an experienced mindfulness instructor. In addition, six studies received the interventions in a homework format, such as an audio CD-ROM and workbooks that guided meditations to support home practice [21], an audio CD-ROM with mindfulness exercises to practice at home [22,23,29,30], and CD-ROMs containing guided sitting meditations [20]. The majority were school-based programs (7/12), whereas one study was carried out in a yoga studio [24]. Regarding the intervention period, most of them lasted eight weeks (8/12), finding that the intervention with the greatest duration lasted 20 weeks [31], whereas the shortest duration was one day (including a 90 min session of mindfulness skills protocol) [25]. Concerning frequency, most studies included at least one session per week (9/12 studies), whereas the duration of the sessions varied across studies, in a range between 40 and 150 min. Finally, four studies [22,24,27,28] included a negative control group (three of these with a wait list control group), and two studies a positive control group, including cooperative activities [32] and pharmacotherapy with risperidone or ritalin [30].

#### 3.3.2. Effects of MBT Interventions on ADHD

Regarding the effect of the interventions on the main ADHD symptoms in children and adolescents, in general, there were significant positive results in eleven out of twelve of the studies (91.6%) [20,21,22,23,24,26,27,28,29,30,31], even though in the study of [25] there were no differences on externalizing symptoms in children after the intervention. With respect to evaluating the main ADHD symptoms, we observed significant improvement across studies on primary outcomes, such as accuracy rate and reaction time [24], hyperactivity and impulsivity [31], as well as on inattention [28].

Secondary outcomes, such as anxiety, shyness, social problems, perfectionism, and impulsivity [31], planning, inhibition, and self-reported emotion dysregulation [27], and depressive symptoms [20] were also significantly improved after MBT interventions. Finally, several specific instruments to measure ADHD symptoms were used (e.g., checklist, questionnaires, visual test, scales, etc.).

### 3.4. Methodological Quality of the Included Studies

In summary, eleven studies were rated “poor”, and only one was rated as “fair” [25] (see Appendix A).

#### 3.4.1. Quality Assessment of Controlled Intervention Studies

We assessed six studies with this tool (i.e., five RCT studies and a clinical trial). Five [24,27,28,30,31] of the six identified controlled studies included a “fatal flaw”, resulting in a “poor” rating. The randomization process was described in all six controlled studies [24,25,27,28,30,31], and all of them correctly described the randomization sequence. Only one RCT study [25] conducted independent recruitment of participants and blinding. All the controlled intervention studies had similar groups at baseline on important characteristics that could affect the outcome of the study. The overall dropout rates from all the studies at endpoint were lower than 20%, as well as the differential dropout rates between intervention and control group, which were over 15%. Adherence to the intervention program was high in all groups, whereas the application of other interventions was not described in any cases. Furthermore, none of the six controlled intervention studies reported an appropriately sized sample necessary for detecting effects with 80% power. Finally, in all studies, all randomized participants were analyzed in the group to which they were originally assigned (per protocol analysis).

#### 3.4.2. Quality Assessment for Before–After Studies (Pre–Post) Studies with No Control Group

We assessed six studies with this tool (i.e., non-random studies without a control group). The six studies [20,21,22,23,26,29] included at least one “fatal flaw”, resulting in a “poor” rating. All studies included the study question and clearly stated the objective, as well as clearly describing eligibility/selection criteria for the study participants. In the six studies, participants were representative of those who were eligible for the intervention, but they did not include a sample size appropriate to provide assurance in their findings. The outcome assessments were detailed, clearly described, valid, reliable, and assessed consistently across all study participants; however, in none of the studies were the people who assessed the outcomes blinded to the participants’ interventions. In all of the studies, the loss to follow-up after baseline was lower than 20%. Regarding statistical methods, only three studies [20,23,29] examined changes in outcome measures between pre–post intervention, providing the *p* values. In addition, in three studies [20,21,26], statistical analysis used the individual-level data to determine effects at the group level. Finally, in three studies [21,23,26], the outcome was measured at multiple times along the MBT intervention. Due to heterogeneity in the measurement of mind–body therapies outcomes (i.e., inattention, hyperactivity, or impulsivity, and related skills), type intervention (i.e., yoga, meditation, mindfulness, relaxation, or zen therapies or programs), and on types of comparators (i.e., no treatment as negative control group or traditional intervention as positive control group), doing a meta-analysis was not possible (see Table 1).

## 4. Discussion

The main findings of this SR were that a significant positive result was found in eleven out of twelve included studies regarding the effect of the MBTs on ADHD symptoms in children and adolescents. In addition, in relation to the quality of the studies, eleven studies were rated “poor” and one was rated as “fair”. A more detailed description of the types of intervention used and the effects produced on the main ADHD symptoms and other related factors are described below.

ADHD persists into adulthood in up to 80% of individuals diagnosed as children [33], so prevention and mitigation of ADHD symptoms are important from an early age. Stimulant medication is the most usual first-line treatment for ADHD for people of all ages. RCTs have proven the effectiveness of stimulants in reducing core symptoms of ADHD in youth [18]. However, a high proportion of young people in these trials were considered nonresponders. Additionally, although stimulant medication is a key component of ADHD treatment, it is necessary to explore the multidimensional effect of other complementary interventions for young people with ADHD, such as nonpharmacological programs, i.e., school-based intervention strategies, psychotherapy, and cognitive–behavioral interventions [34]. Consequently, the positive effects of physical exercise have been described on the same catecholaminergic system targeted by stimulant medications for ADHD [35,36]. Additionally, pharmacotherapy has been related to side effects such as poor tolerance, no response to treatment, and even dependence [10]. Thus, physical exercise could be considered a helper to decrease behavioral problems that influence the learning and academic processes in children with ADHD [10].

As a form of exercise, MBTs, including tai chi, meditation, mindfulness, and yoga, have been put in the crosshairs of the scientific community to analyze in depth the effectiveness of these widely used practices. Accordingly, the National Center for Complementary and Alternative Medicine designates MBTs as a top research priority [6]. According to Morgan et al., MBTs such as tai chi or yoga are multidimensional behavioral therapies that integrate several aspects of physical exercise, such as light to moderate aerobic physical activity, deep breathing, balance, and meditation to encourage stress reduction and relaxation, which could potentially impact ADHD symptoms. Moreover, it has been demonstrated that meditation is a more integrative program for stress reduction, regulating emotional, and affective responses to stress. A recent review [37] with 12 preliminary studies of yoga in schools for children, established that yoga interventions applied positive effects on factors such as emotional balance, attentional control, cognitive efficiency, anxiety, negative thought patterns, emotional and physical arousal, reactivity, and negative behavior. Additional research in children also showed the positive effects of school-based yoga programs on several aspects of mental health, such as concentration, attention, anxiety, stress, mood, resilience, emotional arousal, self-esteem, and coping frequency [38]. However, to date few studies were carried out with MBTs, such as yoga or mindfulness, in children and adolescents with ADHD.

In the current SR, we included twelve studies based on MBTs in children and adolescents with diagnoses of ADHD. Nine developed mindfulness treatments [20,21,22,23,25,27,28,29,30], three conducted yoga programs [24,26,31], and none of these included others such as Zen, tai chi, or specific meditation therapies. Most of the included studies (11/12) displayed significant positive results on ADHD symptoms, where only the study of Gershy et al. [25] did not show significant positive results regarding ADHD symptoms after the intervention. A possible explanation for this lack of effect on ADHD symptoms could be that the intervention period was only one day, including a session of 90 min of mindfulness skills protocol. In this sense, evidence-based interventions often involved 7 to 12 weekly sessions, demonstrating improved child behavior and parent satisfaction [39,40], in addition three to four weeks on stable medication is the minimum length of treatment efficacy in subjects with ADHD [41]. Accordingly, the present SR displayed that most of the included studies (8/12) carried out an intervention for at least eight weeks, including a study with a duration of 20 weeks [31]. So, future intervention studies should include intensive training lasting at least seven to eight weeks of intervention. For example, in two case studies of children with ADHD, a significant decrease in compliance of the child was described after an intensive mindfulness program including 12 sessions (twice weekly) [42]. Future research must establish the optimal frequency, duration, and intensity of intervention programs to lead to effective interventions.

In the study of [25], no differences in child externalizing symptoms were found, which decreased significantly in both groups after intervention. However, mothers’ negative feelings, escalating behaviors, and capacity for emotional regulation improved significantly at the end of the intervention. Moreover, fathers in the mindfulness condition reported greater improvement in the capacity for emotional regulation, reduced negative feelings, and reduced parental submission compared with fathers in the parent training condition. Thus, it is possible that interventions focused only on youth may be more effective than those that include both young people and parents. Contrary to this hypothesis, effective behavioral therapies include parent and youth training, with classroom management, peer interventions, or combinations of these interventions [43]. It seems that parent training improves their understanding of ADHD, behavioral problems, and child development, helping them to use strategies such as praise and rewards for targeted behaviors. Additionally, disruptive child behaviors are decreased. Thereby, MBTs involving both parents and youths seem to have a positive influence on family relationships, further improving ADHD symptoms in children and adolescents, as shown by three of the studies included in the present review [21,22,28]. Additionally, it is important to take into account that ADHD is a neurodevelopmental disorder, which could remit and even transform some symptoms and deficits into adaptive behaviors or even lead a negative trajectory. The patient’s compliance and feeling regarding the medical therapy [44,45,46] and an optimal medical and multicomponent treatment is the key to ensure positive effects.

Concerning study quality, even though we included four RCT studies and a clinical trial, most of the studies (11/12) were rated as “poor” quality, while only the study of [25] was rated as “fair” quality. Again, we hypothesized that the reason for the difference between studies could be the duration of the intervention (i.e., one day of a 90 min session of mindfulness skills protocol), since the short duration (one day) in the intervention could avoid several “fatal flaws” in the design or development of the study (e.g., blinding of participants or providers to the treatment group assignments, etc.).

Another important criteria that could influence the quality assessment differences between the study of Gershy et al. and the rest of the controlled intervention studies could be the dropout rate, since previous literature has shown that there is usually an important prevalence of dropouts in exercise interventions [47]. However, in the current SR, all of controlled intervention studies met both criteria for not having a “fatal flaw”, i.e., overall dropout rate from the study at endpoint (20% or lower) and differential dropout rate (between treatment groups) at endpoint (15% or lower). In this respect, it seems that lower dropout was observed in sports interventions compared with structured aerobic exercise or yoga (*p* = 0.049), so in order to increase participation, and therefore health benefits, MBT interventions must offer a multidisciplinary approach for children and adolescents with ADHD [47].

In addition to the satisfactory dropout rate across the studies, most of the controlled intervention studies had a high adherence to the intervention programmes (4/6). However, regarding before–after studies with no control group, only the studies of Zylowska et al. [20] and Zhang et al. [29] reported a high rate of adherence, with 78% and 91%, respectively. Regarding adherence correlates, family problems was the main reason for nonadherence among children to yoga practice at home post-discharge [26]. The high percentage of adherence, as well as the low dropout rate could be due to different reasons. First, we hypothesized that there was high satisfaction among the participants who enjoyed MBTs. For instance, Zylowska and colleagues (2008) conducted an open trial of a mindfulness training program where high satisfaction ratings from adolescents were reported. Another possible reason that could explain these two positive aspects in the quality of interventions, i.e., dropout and adherence rates, could be that the sample size was small in all of the studies (12/12), allowing greater control and individual monitoring of the participants. However, an insufficiently large sample size was one of the most usual “fatal flaws” across the studies.

In the controlled intervention studies, none reported a sufficiently large sample size to be able to detect a difference in the main outcome between groups with at least 80% power, as well as in the before–after studies with no control group (6/6), in which the sample size was not sufficiently large to provide confidence in the findings. Thus, small sample sizes in these studies precludes the generalizability of the findings, resulting in a poor quality of the study. Therefore, although most studies reported positive benefits on ADHD symptoms of children and adolescents after completing MBTs, their quality was rated mostly as poor, so future intervention studies should include larger samples as well as randomization and control groups, in order to clarify these findings.

Results of our review study must be viewed in the light of methodological limitations. Overall, the “poor” and “fair” quality ratings for the majority of studies makes it difficult to conclude with consistency regarding future intervention strategies, mainly because risk of bias in the study design critically decreases the confidence in the rationality of the results. Albeit, some aspects must be considered in cases of exercise intervention studies. First, it is possible to find gaps when assessing the quality of the studies using the NIH tools, since these were designed principally for clinical trial studies. For example, the blinding element could be considered inapplicable because it is practically impossible to blind participants who receive MBTs such as yoga or mindfulness. Second, some studies did not specify the intensity/frequency of the intervention, as well as quality criteria such as adherence or in-depth details of the statistical methods. Third, most studies did not report if the participants were involved or stopped some medical or psychological treatment for their ADHD symptoms, or if they performed some additional physical activity, sport, or exercise. Neither were the participants analyzed according to the subtype of ADHD diagnosis. Fourth, the quality of included studies ranked mainly as poor; however, we did not exclude any because of the small number of articles selected. As a strength, a rigorous and strict review process was performed for selecting the studies and extracting the data. Additionally, two valid instruments to evaluate the studies quality were used. Finally, we have not included in this manuscript an SR with a quantitative analysis (meta-analysis) due to the considerable conceptual heterogeneity in the studies (systematic differences in study design, patient populations, interventions or co-interventions, clinical heterogeneity, including duration of the study, the dose of the intervention, and the evaluation of the results). In addition, we observed that the studies were systematically different from each other, so the quantitative synthesis would not be generalizable and applicable to clinical practice. In this context, we recognize and explain the heterogeneity in inclusion studies, particularly from a qualitative perspective, which is a general sense of what all studies say, and is absolutely crucial, especially when analyzing controlled intervention and before–after studies.

## 5. Conclusions

Most of the intervention studies conducting MBTs in children and adolescents with ADHD indicated positive results for improving ADHD symptoms. The majority of them were eight week school-based programs based on mindfulness treatment, guided by a certified instructor, with a frequency of one session per week, and with duration in a range between 40 and 150 min. However, the studies were identified as being of poor quality. Thus, further studies should be more rigorous to improve the consistency and generalizability of their findings. They must include the use of representative samples with greater statistical power and randomized controlled trial designs. Finally, further research should include behavioral interventions such as MBTs in conjunction with pharmacological treatment.

## Figures and Tables

**Figure 1 medicina-55-00325-f001:**
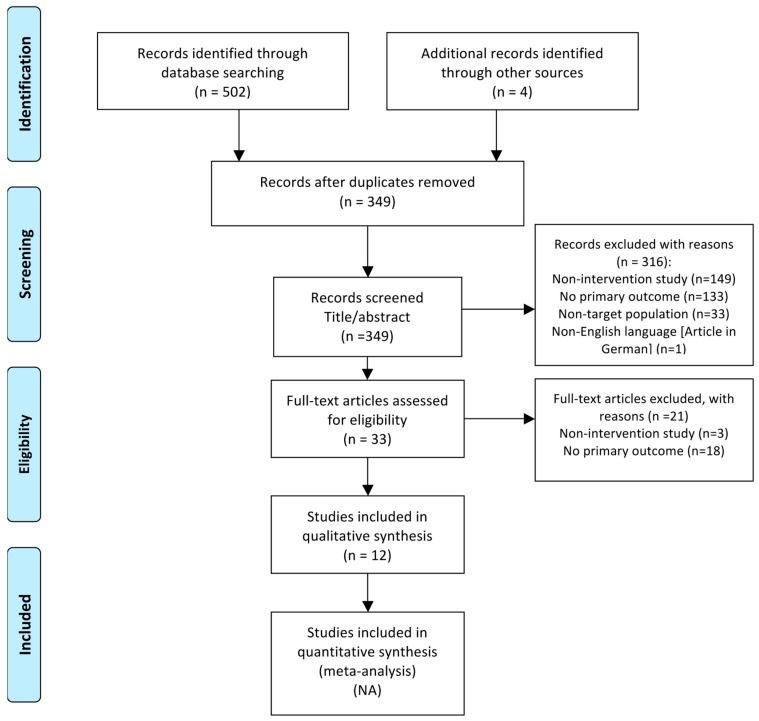
PRISMA flow diagram of included studies on Mind–Body Therapies for the improvement of attention-deficit/hyperactivity disorder symptoms in children and adolescents.

**Table 1 medicina-55-00325-t001:** Characteristics of interventions of mind–body therapies for the improvement of ADHD symptoms (n = 12).

Author, Year and Country	Sample and Age	Intervention Design	Type of Intervention	Duration and (Frequency)	Outcomes and Tests	Main Results	Quality ^#^
[24] Taiwan	49 (8–12 yr)	Non-randomized controlled trial	Yoga exercise group versus control group (negative)	8 weeks (twice per week, 40 min per session)	**Primary outcomes:** Selective and sustained attention through Visual Pursuit Tests, and selective attention by the Determination test. In both tests, the attention outcomes were evaluated through the accuracy rate and reaction time (RT).	After the intervention, significant improvements in accuracy rate and reaction time of the two tests were observed the yoga exercise group compared with the control group. In the visual pursuit test, the exercise group reported an increased accuracy rate after the yoga intervention (*t*_23_ = −2.12, *p* = 0.045, d = −0.69), and a faster RT at the post-test than the control group (*t*_47_ = −4.18, *p* < 0.001, d = −1.20). For determination test, the yoga exercise group reported an increased response accuracy (*t*_23_ = 5.78, *p* < 0.001, d = 1.22) and a faster RT at the post-test than the control group (*t*_47_ = −4.26, *p* < 0.001, d = 1.25)	poor
[25] Israel	57 (6–15 yr)	Non-randomized controlled trial	Mindfulness Training versus non-violent resistance parent training	1 day (90 min mindfulness skills protocol)	**Primary outcomes:** Child externalizing symptoms were evaluated through the Child Behavior Checklist questionnaires and an intake session for a clinical professional pchycologist	After intervention, parents in treatment conditions reported a significant reduction in child externalizing symptoms on the CBCL externalizing scale (ß = −4.73 (1.56), *t* = −3.04, *p* = 0.002); change over time represented 51% of the within-person variance.	fair
[26] India	9 (5–16 yr)	Non-randomized controlled trial	Yoga exercise group	In-patient stay in psychiatry unit. (1 h session with at least six sessions)	**Primary outcomes:** ADHD symptoms through ADHD rating scale-IV (ADHD-RS), Conners’ abbreviated rating scale (CARS) and clinical global impression (CGI) severity, were evaluated at the beginning of study, at discharge, and subsequently at the end of 1st, 2nd, and 3rd month by a research associate not involved in yoga instruction.	A reduction in scores from baseline to discharge in all the scales was observed (*p* = 0.014 on CARS and *p* = 0.021 on ADHD-RS and 0.004 on CGI). There was no significant reduction in the scores later on.	poor
[21] Canada	18 (13–18 yr)	Non-randomized controlled trial	Mindfulness-based cognitive therapy (MBCT)	8 weeks (1.5 h session each week)	**Primary outcomes:** Screen of ADHD symptoms (inattention, hyperactivity/impulsivity, learning problems, aggression, oppositionality, and relationships with others) were analyzed by self-report (Conners 3-SR) scales. Secondary outcomes: Revised Child Anxiety and Depression Scale—Youth (RCADS) at four time points (1 pre and 3 post)	Reductions in the adolescents’ inattention (near significant *p* = 0.07; d = 0.62), conduct problems (*p* < 0.05; d = 0.70), and peer relations problems (*p* < 0.05) after the MBCT intervention, according to parental report.	poor
[31] Australia	18 (8–13 yr)	Randomized crossover design	Yoga versus control group (positive)	20 weeks (1 h program)	**Primary outcomes:** Screen of ADHD symptoms (inattention, hyperactivity/impulsivity, learning problems, aggression, oppositionality, and relationships with others) were analyzed by the Conners’ Parent and Teacher Rating Scales. Attention and impulsivity were evaluated by the Test of Variables of Attention (TOVA), and hyperactivity was assesed by the Motion Logger Actigraph.	Regarding the Conners’ Parent and Teacher Rating scales, there were significant effects for the yoga group on five subscales of the CPRS: Oppositional (*p* = 0.003, Cohen’s d = 0.77); Global Index Emotional Lability (*p* = 0.001, Cohen’s d = 0.79); Global IndexTotal (*p* = 0.001; Cohen’s d = 0.73); Global Index Restless/Impulsive (*p* = 0.008, Cohen’s d = 0.73) and ADHD Index (p = 0.019, Cohen’s d = 0.29). Days of home practice were positively associated with three factors—TOVA Response Time Variability (r = 0.628, *p* = 0.038), TOVA/ADHD score (r = 0.648, *p* = 0.05), and CTRS–R:L Global Emotional Lability (r = −0.628, *p* = 0.05)	poor
[27] Iran	30 (13–15 yr)	Randomized crossover design	Mindfulness treatment group versus waitlist control group (negative)	8 weeks (1 session/1.5-h)	**Primary outcomes:** Executive functioning was assesed by the following test: Continuous performance test (CPT for continuous attention dysfunction and response inhibition, Digit Span (Forward and Backward) subtest and Letter–Number **Secondary outcomes:** Sequencing subtest of working memory index of Wechsler Intelligence Scale for Children—Fourth Edition (WISC-IV), Stroop Word–Colour Interference Test for inhibition, and Tower of London test.	For executive functions: Treatment group had a significantly higher inhibition than the control group at post-test (F(1, 27) = 7.58, *p* = 0.01, partial η2 = 0.22). Planning of the treatment group was significantly different than the control group (F(1, 27) = 4.88, *p* = 0.04; partial η2 = 0.15). For Emotion dysregulation, the treatment group had lower emotion dysregulation (F(1, 27) = 6.41, *p* = 0.02, partial η2 = 0.19), Nonacceptance of Emotion Responses (F(1, 27) = 9.67, *p* = 0.004; partial η2 = 0.26), and Impulse Control Difficulties (F(1,27) = 7.97, *p* = 0.009; partial η2 = 0.26), in comparison with the control group.	poor
[28] China	100 (5–7 yr)	Randomized Control Trial	Family-based mindfulness intervention versus waitlist control group (negative)	6 weeks (8 sessions/child program)	**Primary outcomes:** Inattention and hyperactivity/impulsivity by SWAN scale, Internalizing and externalizing problems by Child Behavior Checklist, Attention by Child Behavior Checklist, ADHD Symptoms by The ADHD Rating Scale [ARS] assesed at pre tests, immediately after the 8 week training, and at 8 week follow-up.	Families from intervention group had greater improvements in children’s ADHD symptoms, with medium effect sizes of −0.60 for inattention and −0.59 for hyperactivit compared with the wait-list control group. with the wait-list control group	poor
[22] Netherlands	22 (8–12 yr)	Non-randomized controlled trial	Mindfulness School-based program and within-group waitlist (negative)	8 weeks (1.5 h sessions)	**Primary outcomes:** Disruptive behavior disorder symptoms through Disruptive Behavior Disorder Rating Scale [DBDRS].	From pre to post test, significant reduction of inattention (ES = 0.80, large ES) and hyperactivity/impulsivity symptoms of the child (ES = 0.56, medium ES) on the parent-rated DBDRS, and a significant reduction of the parental inattention and hyperactivity/impulsivity symptoms on the ARS, with small effect sizes (ES = 0.36 and 0.48, respectively). From pre to follow-up, again, inattention and hyperactivity/impulsivity showed significant reductions with small to large effect sizes in both parent (ES = 0.34/0.50, respectively) and child (ES = 0.80/0.59, respectively).	poor
[20] EEUU	8 (mean age = 15.6 (1.1) yr)	Non-randomized controlled trial	Mindfulness	8 week program (once-per-week evening sessions lasting 2.5 h and daily at-home practice)	**Primary outcomes:** ADHD symptoms by the SNAP-IV scale (adolescents). Attentional conflict by Stroop task. Attentional set-shifting and inhibition by the Trail Making Test. Attention through the Attention Network Test. Working memory by the Digit Span. **Secondary outcomes:** Intelligence Scale–Revised or Wechsler Intelligence Scale. Self-report of anxiety and depression was assessed via the Child Depression Inventory and the Revised Children’s Manifest Anxiety Scale (RCMAS) for adolescents. Verbal IQ by Wechsler Intelligence Scale for Children—Third Edition,	Regarding ADHD symptoms, significnatly reductions were observed for inattentive and combined (*p* < 0.001). For cognitive measures, a significant decrease was observed in Attention Network Test conflict, Trails A, and Trails B (*p* < 0.001), and a significant increase was observed in Stroop color word. Slight changes were observed for depression and anxiety.	poor
[29] China	11 (8–12 yr)	Non-randomized controlled trial	Mindfulness	8 weeks (90 min group sessions)	**Primary outcomes:** Childrens’ attention and related problems by The Conners’ Continuous Performance Test 3rd Edition (CPT 3) and the Test of Everyday Attention for Children (TEA-Ch) for selective attention, sustained attention, and attention control/switching. **Secondary outcomes:** Parents’ perception of disruptive behavior by the Eyberg Child Behavior Inventory (ECBI)	Positive results with large effect sizes only occurred on the objective attention tests: time per target (*p* = 0.0003; ES = 1.53, large ES), the attention score (*p* = 0.001; ES = 1.35, large ES) of the Sky Search test, and Map Mission test (*p* = 0.0001; ES = 1.27, large ES) of TEA-Ch. For omissions, T score improved from 87.9 (6.9) to 58.4 (16.9), *p* = 0.0003; ES = 2.29, large ES) in the computerized CPT 3 test, which indicates better attention. No statistically significant changes were seen in other measures of the Test of Everyday Attention for Children (TEA-Ch).	poor
[23] Netherlands	10 (11 to 15 yr)	Non-randomized controlled trial	Mindfulness	8 weeks (1.5 h sessions.)	**Primary outcomes:** Attention and behavioral problems Youth Self Report (YSR) and the Child Behavior Checklist (CBCL), **Secondary outcomes:** Executive Function The Behavior Rating Inventory of Executive Function (BRIEF) Mindful awareness. Mindful Attention and Awareness Scale (MAAS)	After mindfulness training, significant reduction in attention problems was reported after training by fathers for attention, (95% CI, −5.35 to 0.046, *p* = 0.09 and in internalizing problems (95% CI, −8.09 to 0.09, *p* = 0.05). Metacognition and behavioral regulation significantly improved after training, reported by tutors (95% CI, −18.94 to 1.39, *p* = 0.08), reported bu fathers (95% CI, −12.97 to 0.40, *p* = 0.06)	poor
[30] Iran	56 (7–12 yr)	Randomized Control Trial	Mindful Parenting Training versus control group (positive)	8 weeks (1 session/90 min)	**Primary outcomes:** Attention deficit, hyperactivity Disorder, attention deficit hyperactivity disorder. **Secondary outcomes:** Swanson, Nolan, and Pelham Parent and Teacher Rating Scale (SNAP-IV)	There were signifficant differences between experimental and control group after the intervention program (postest) in all measured variables: Attention deficit (*p* = 0.001), hyperactivity (*p* = 0.04), and attention deficit hyperactivity disorder (*p* = 0.04)	Poor

Included: strongly related to the primary outcome and study population (children and adolescents), ANT: Attention Network Test, ADHDRS: Rating Scale of ADHD, CBCL: Child Behavior Checklist, CPRS: Comprehensive Psychopathological Rating Scale, TOVA: Test of Variables of Attention, MAAS: Mindful Attention and Awareness Scale, AGN: Affective Go/No-Go test, MAAS: Mindful Attention and Awareness Scale, AGN: Affective Go/No-Go test. **^#^** NIH Quality Assessment Tools for (1) Quality Assessment of Controlled Intervention Studies (i.e., RCT and clinical studies); and (2) Quality Assessment Tool for Before–After (Pre–Post) Studies with No Control Group.

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
