# Peer review of "Interventions Based on Mind–Body Therapies for the Improvement of Attention-Deficit/Hyperactivity Disorder Symptoms in Youth: A Systematic Review"

_medicina, 2019, doi:10.3390/medicina55070325_

Round 1

Reviewer 1 Report

The interventions based on mind-body therapies are among "friendly" interventions for individuals with ADHD and analysis of their impact on ADHD symptoms dynamics is an important topic in this field. Thus the present systematic review (SR) has a potential to provide an updated knowledge of this issue. The authors follow the required PRISMA procedure although the SR has not been registered on PROSPERO. The following comments/questions arise:

1) The main aim for the SR in its present formulation is a bit ambiguous, it should be further specified whether SR aims at analyzing interventions themselves (their nature, scope, etc) or their impact on ADHD symptoms dynamics;

2) there is lack of information on the records excluded at screening step in PRISMA process: what happened to the 252 records that were not screened? Also, there is mismatch between numbers of the studies described in Supplementary Table 2 and information given (p. 5, lines 162-167). Studies that were exluded at any step should not appear on Reference list.

3) Reasons for exclusion should be indicated at screening and eligibility steps in PRISMA figure (Supplementary list not required);

4) Authors state that "the main outcome of this study was ADHD symptoms, such as inattention, hyperactivity, or impulsivity, and related skills, which are attenuated with ADHD diagnostics (e.g. accuracy rate, reaction time, etc.)" (p. 2, lines 87-89). However, description of results from the main outcome is much wider, including emotional, personality, parental and other variables (Section 3.4 & lines 290-295). Authors should reconsider the main outcome description and structure better its description.

Author Response

Response to the Reviewers’ Comments

Dear Editor-in Chief

Please, find a new revision of our manuscript “Interventions based on mind-body
therapies for the improvement of Attention-Deficit/ Hyperactivity Disorder symptoms in
youth: A systematic review”. We would like to thank the Editorial Committee and the
Reviewers for their thoughtful and constructive comments. We have considered all the
suggestions, and have incorporated them into the revised manuscript. We believe our
manuscript is stronger because of these revisions. An itemized point-by-point response
to the Reviewers’ comments is presented below.

Authors response to Reviewer 1

1) The main aim for the SR in its present formulation is a bit ambiguous, it should
be further specified whether SR aims at analyzing interventions themselves
(their nature, scope, etc) or their impact on ADHD symptoms dynamics.

• Response: Thank you for this relevant and constructive comment. The
objective of this SR is as follows: Analyze the interventions based on MBT
aimed to improve the main ADHD symptoms in children and adolescents.
Following the recommendations of PRISMA methodology for SR based on
interventions studies, first, a description of the characteristics of the studies
should be made (sample, country, etc.) as well as a detailed description of
the interventions carried out. In addition, a description of the effects of these
interventions on the main and secondary outcomes (please, see in table 1),
and finally, a quality assessment regarding the study design. All these phases
have been carried out in the present SR.

However, we totally agree with reviewer's comment, since in the last
manuscript version, the titles chosen in the sub-sections of the results about
the description of the outcomes of the studies analyzed in this SR led to
confusion. Therefore, we have improved all of them, as well as, we have
homogenized this idea through the manuscript changing the objective, the
first paragraph of the discussion and the conclusion.

New results sections:

3. Results sections
3.1. Study selection.
3.2. Characteristics of the included studies.
3.3. Analysis of the MBT intervention and its effects on ADHD
outcomes
3.3.1. MBT interventions
3.3.2. Effects of MBT interventions on ADHD

New first paragraph of discussion:

The main findings of this SR were that a significant positive result was found
in eleven out of twelve included studies regarding the effect of the MBTs on
main ADHD symptoms in children and adolescents. In addition, in relation
to the quality of the studies, eleven studies were rated “poor” and one was
rated as “fair”. A more detailed description of the types of intervention used
and the effects produced on the main symptoms of ADHD and other related
factors are discussed below.

New conclusion:

Most of the intervention studies conducting MBTs in children and
adolescents with ADHD indicated positive results for improving the main
ADHD symptoms. The majority of them were 8-week school-based
programmes based on mindfulness treatment guided by a certified instructor
with a frequency of one session per week with duration in a range between
40 and 150 minutes. However, the studies were identified as being of poor
quality. Thus, further studies should be more rigorous to improve the
consistency and generalizability of their findings. They must include the use
of representative samples with greater statistical power and randomised
controlled trial designs. Finally, further research should include behavioural
interventions such as MBTs in conjunction with pharmacological treatment.
Again, thanks for this comment; since we believe that thanks to your
appreciation, the general comprehension of the manuscript has improved.

2) There is lack of information on the records excluded at screening step in
PRISMA process: what happened to the 252 records that were not screened?
Also, there is mismatch between numbers of the studies described in
Supplementary Table 2 and information given (p. 5, lines 162-167). Studies that
were excluded at any step should not appear on Reference list.

• Response: Thanks, it was a mistake. We have reorganized the flow diagram,
specifying the reasons for exclusion in the title / abstract phase (screening) as
well as in the full-text phase (elegibility). In addition, we have deleted the
supplementary table 2, which would now be redundant, and we have also
deleted the references of the studies excluded from references list. Finally,
the paragraph of the text (p.5, 3.1 Study selection) has been changed taking
into account new changes in the flow diagram. Now it is presented as
follows;

In the first stage of the search strategy, a total of (n=506) articles were identified. In
the second stage, following the removal of duplicates (n=157), a total of (n=349)
were screened by title/abstract. Then, 316 potentially relevant articles were
excluded with reasons (details summarized in figure 1). In the third stage, full-text
articles were reviewed in depth and (n=21) studies were excluded with reasons
(details summarized in figure 1). Finally, only (n=12) studies based on MBT
interventions met the inclusion and exclusion criteria and were included in the
final analysis.

3) Reasons for exclusion should be indicated at screening and eligibility steps in
PRISMA figure (Supplementary list not required).

• Response: Thank you for this comment. As we have noticed, we included
the reasons for exclusion in the flow diagram for both, screening and
eligibility steps in the PRISMA figure. In addition, the supplementary table 2
has been removed.

4) Authors state that "the main outcome of this study was ADHD symptoms, such as
inattention, hyperactivity, or impulsivity, and related skills, which are attenuated with
ADHD diagnostics (e.g. accuracy rate, reaction time, etc.)" (p. 2, lines 87-89).
However, description of results from the main outcome is much wider, including
emotional, personality, parental and other variables (Section 3.4 & lines 290-295). Authors should reconsider the main outcome description and structure better its description.

• Response: Thank you for this constructive comment. We agree with your
appreciation and for that reason, we have rewritten the outcomes of the
analyzed studies included in this SR, including clearly the main and
secondary outcomes, since we consider that reflexing about all factors is
important to broadly address the conclusions of the SR. Additionally, as we
have noted previously, we have improved this issue in the abstract,
discussion and conclusion sections.

Reviewer 2 Report

this is an interesting paper about systematic review of Attention-deficit/hyperactivity disorder syntomps in youth.

Some criticism are present:

1 during the description of the interventions also the types of comparators (i.e. no treatment as negative control group or traditional intervention as positive control group)should be described.

2.Any indication about the possibility of performing a meta-analysis of data extracted from the included studies was gave. The differences of effectiveness between interventions and control groups were never supported with statistical parameters such as risk ratio (95% CI) for dichotomous outcomes or mean difference (MD) for continuous data. The statistical significance of effectiveness differences could be also expressed even only in terms of p-value

3.In the flow-chart diagram the excluded studies (full texts) with reason could be classified in accordance to the different study design (i.e. n=3 Reviews, n=2 Observational studies, n= 3 Case series.....)

4 the primary and secondary outcomes could be reported a distinct paragraph of the methods section of this study 

5 Some important considerations should be added about clinical relevance of ADHD in term of patients compliance and difficult of therapies. In this context some consideration about medical aspects must be added in Introduction or Conclusone session.

I suggest i these scientific paper to insert in reference section about this statement :

-Cianetti S,Lombardo G, Lupatelli E, Pagano S, Abraha I, Montedori A, Caruso S , Gatto R, e Giorgio S, Salvato R. Dental fear, anxiety among children and adolescents. A systematic review. Our J Paediatr Dent. 2017 Jun;18(2):121-130.

-Paglia L, Gallus S, de Giorgio S, Cianetti S, Lupatelli E, Lombardo G, Montedori A, Eusebi P, Gatto R, Caruso S. Reliability and validity of the ital Version of the children's fear survey schedule-dental sub scale and modified child dental anxiety scale. Our J Paediatr Dent . 2017 Dec; 18(4):305-312

Author Response

Response to the Reviewers’ Comments

Dear Editor-in Chief

Please, find a new revision of our manuscript “Interventions based on mind-body
therapies for the improvement of Attention-Deficit/ Hyperactivity Disorder symptoms in
youth: A systematic review”. We would like to thank the Editorial Committee and the
Reviewers for their thoughtful and constructive comments. We have considered all the
suggestions, and have incorporated them into the revised manuscript. We believe our
manuscript is stronger because of these revisions. An itemized point-by-point response
to the Reviewers’ comments is presented below.

Authors response to Reviewer 2

This is an interesting paper about systematic review of Attention-deficit/hyperactivity
disorder symptoms in youth. Some criticism are present:

1) During the description of the interventions also the types of comparators (i.e. no
treatment as negative control group or traditional intervention as positive control
group) should be described.

• Response: Thank you for the positive feedback regarding our study, as well
as this constructive comment. We have included the required information,
i.e., types of comparators in Table 1, highlighting if the control group is
negative or positive. Additionally, we have included a new sentence
deepening in these details (please, see 3.2.1 MBT interventions section):
Finally, four studies (22,24,27,28) included a negative control group (three
of these with a wailt-list control group) and two studies a positive control
group, including cooperative activities (32) and pharmacotherapy with
risperidone or Ritalin (30).

2) Any indication about the possibility of performing a meta-analysis of data
extract rom the included studies was gave. The differences of effectiveness
between interventions and control groups were never supported with statistical
parameters such as risk ratio (95% CI) for dichotomous outcomes or mean
difference (MD) for continuous data. The statistical significance of effectiveness
differences could be also expressed even only in terms of p-value.

• Response: We agree with the reviewer comment and therefore, we have
included a new sentence in the manuscript in the end of the result section as
follow:

Due to heterogeneity in the measurement of mind-body therapies outcomes
(i.e.: inattention, hyperactivity, or impulsivity, and related skills), type
intervention (i.e. yoga, meditation, mindfulness, relaxation, or zen therapies
or programmes), and on types of comparators (i.e. no treatment as negative
control group or traditional intervention as positive control group), doing a
meta-analysis was not possible (see Table 1).

In addition, the end of the discussion section we also declare this issue:
Finally, we have not included in this manuscript a SR with a quantitative
analysis (meta-analysis) due to the considerable conceptual heterogeneity in
the studies (systematic differences in study design, patient populations,
interventions or co-interventions, clinical heterogeneity, including duration
of the study, the dose of the intervention and the evaluation of the results); In
addition, we observed that the studies were systematically different from
each other, so the quantitative synthesis would not be generalizable and
applicable to clinical practice. In this context, we recognize and explain the
heterogeneity in inclusion studies, particularly from a qualitative
perspective, which is a general sense of what all studies say, and is
absolutely crucial, especially when analyzing Controlled Intervention and
Before-After Studies.

3) In the flow-chart diagram the excluded studies (full texts) with reason could be
classified in accordance to the different study design (i.e. n=3 Reviews, n=2
Observational studies, n= 3 Case series.....)

• Response: Thanks, we have reorganized the flow diagram, specifying the
reasons for exclusion in the title / abstract phase (screening) as well as in the
full-text phase (elegibility). In addition, we have deleted the supplementary
table 2, which would now be redundant, and we have also deleted the
references of the studies excluded from references list. Finally, the paragraph
of the text (p.5, 3.1 Study selection) has been changed taking into account
new changes in the flow diagram. Now it is presented as follows;
In the first stage of the search strategy, a total of (n=506) articles were identified. In
the second stage, following the removal of duplicates (n=157), a total of (n=349)
were screened by title/abstract. Then, 316 potentially relevant articles were
excluded with reasons (details summarized in figure 1). In the third stage, full-text
articles were reviewed in depth and (n=21) studies were excluded with reasons
(details summarized in figure 1). Finally, only (n=12) studies based on MBT
interventions met the inclusion and exclusion criteria and were included in the
final analysis.

4) The primary and secondary outcomes could be reported a distinct paragraph of
the methods section of this study.

• Response: Thank you for this comment. We agree with your appreciation
and for that reason, we have rewritten the outcomes of the analyzed studies
included in this SR as main and secondary outcomes, since we consider that
reflexing about all factors is important to broadly address the conclusions of
the review. Now this topic is clearly detected in methods, results, discussion
sections and and table 1 (please see new inclusion).

5) Some important considerations should be added about clinical relevance of
ADHD in term of patients compliance and difficult of therapies. In this context
some consideration about medical aspects must be added in Introduction or
Conclusion session.
• Response: Thank you for this interesting suggestion. We have included this
sentence and the 3 new references about this issue:
It is important to take into account that ADHD is a neurodevelopmental
disorder which could remit and even transform some symptoms and deficits
into adaptive behaviours or even lead a negative trajectory. The patient's
compliance and feeling regarding the medical therapy (Cianetti 2017, Paglia
L, 2017 and Franke 2018), and an optimal medical and multicomponent
treatment is the key to ensure positive effects.

New references:
Cianetti S,Lombardo G, Lupatelli E, Pagano S, Abraha I, Montedori A, Caruso
S , Gatto R, e Giorgio S, Salvato R. Dental fear, anxiety among children and
adolescents. A systematic review. Our J Paediatr Dent. 2017 Jun;18(2):121-
130.
Paglia L, Gallus S, de Giorgio S, Cianetti S, Lupatelli E, Lombardo G,
Montedori A, Eusebi P, Gatto R, Caruso S. Reliability and validity of the ital
Version of the children's fear survey schedule-dental sub scale and modified
child dental anxiety scale. Our J Paediatr Dent . 2017 Dec; 18(4):305-312
Franke, B., Michelini, G., Asherson, P., Banaschewski, T., Bilbow, A.,
Buitelaar, J. K., ... & Kuntsi, J. (2018). Live fast, die young? A review on the
developmental trajectories of ADHD across the lifespan. European Neuropsychopharmacology.
6) I suggest these scientific paper to insert in reference section about this statement:
Cianetti S,Lombardo G, Lupatelli E, Pagano S, Abraha I, Montedori A, Caruso S , Gatto R, e Giorgio
S, Salvato R. Dental fear, anxiety among children and adolescents. A systematic review. Our J
Paediatr Dent. 2017 Jun;18(2):121-130.
Paglia L, Gallus S, de Giorgio S, Cianetti S, Lupatelli E, Lombardo G, Montedori A, Eusebi P, Gatto
R, Caruso S. Reliability and validity of the ital Version of the children's fear survey schedule-dental
sub scale and modified child dental anxiety scale. Our J Paediatr Dent . 2017 Dec; 18(4):305-312
o Response: Thank you for your suggestion, the references have been
included.

Round 2

Reviewer 1 Report

Authors have addressed the comment adequately to improve the manuscript. Thank you.